# Evaluating Measurement Properties of the Adapted Interprofessional Collaboration Scale through Rasch Analysis

**DOI:** 10.3390/healthcare10102007

**Published:** 2022-10-12

**Authors:** Heike Wieser, Maria Mischo-Kelling, Luisa Cavada, Lukas Lochner, Verena Fink, Carla Naletto, Fabio Vittadello

**Affiliations:** 1Research Unit, College of Health-Care Professions, 39100 Bolzano, Italy; 2Faculty of Social Work, Health and Nursing, University of Applied Sciences Ravensburg-Weingarten, 88250 Weingarten, Germany; 3Postgraduate Courses Area, College of Health-Care Professions, 39100 Bolzano, Italy; 4Teaching Support Office, College of Health-Care Professions, 39100 Bolzano, Italy; 5Nursing Directorate, South Tyrolean Healthcare Trust, 39100 Bolzano, Italy; 6Department of Physiotherapy, College of Health-Care Professions, 39100 Bolzano, Italy; 7Explora—Research and Statistical Analysis, 35010 Padova, Italy

**Keywords:** interprofessional collaboration, Rasch analysis, IPC scale, nurses, instrument validation

## Abstract

This study aimed to evaluate the validity and reliability of the German and Italian versions of the Interprofessional Collaboration Scale (IPC scale) by applying a Rasch analysis. Data were gathered from 1182 nurses participating in a cross-sectional study in northern Italy. The scale demonstrated good reliability (Cronbach’s alpha = 0.92). Item polarity of all 13 items was positive, indicating good construct validity. However, revising one item would further improve the validity of the scale. Item stability was confirmed for work experience, workplace, age range, the language version, and gender. The analysis, applying a non-classical test theory, confirmed that the IPC scale is a valid and reliable instrument to measure interprofessional collaboration between nurses and physicians.

## 1. Introduction

Interprofessional collaboration (IPC) is defined as the collaboration of two or more healthcare workers with different professional backgrounds, and it is considered to be a precondition for safe and person-centered healthcare provision [1,2]. According to Schot et al. [3], IPC is co-created by healthcare workers so that not only policy and management must contribute to it but also that every health professional can actively play their part to facilitate a collaborative practice. This can occur in looser (networks) and tighter (team) forms, as described by Xyrichis and Reeves [4,5]. In recent years, there have been many efforts to study IPC.

The availability of valid and reliable scales measuring IPC with its different components, such as communication, accommodation, and isolation, for more than two different healthcare professions is, according to Walters et al., still scarce [6]. Researchers from the Joanna Briggs Institute in Adelaide conducted a systematic review of the psychometric properties of instruments measuring collaboration. They created a list with evaluated instruments serving as a valuable source for researchers and healthcare practitioners. They evaluated and compared 12 available instruments to assess collaboration within the healthcare sector, considering the already published psychometric properties of these instruments. The Interprofessional Collaboration Scale, hereafter the IPC scale, of Kenaszchuk et al. [7] was the only available instrument measuring collaboration in multi-rater target groups. It was rated as an instrument with good content validity and moderate internal consistency, structural and criterion validity, with the potential for measuring interprofessional collaboration in different professional groups [6]. In addition, the scoping review from Peltonen et al. [8], including 29 instruments, emphasized that the scale developed by Kenaszchuk et al. [7] was among the rare studies that reported on construct and criterion validity as well as reliability testing.

Kenaszchuk et al. [7] developed the tool in 2010. In their article “Validity and reliability of a multiple-group measurement scale for interprofessional collaboration”, they outlined the rationale and need to construct a new measurement scale for multiple raters, as none was available at that time. The authors described all phases of the construction, adaptation, and validation process of the newly created IPC scale. They used and adapted two subscales of the Nurse Opinion Questionnaire (NOQ) to develop their new IPC scale. They applied exploratory and confirmatory factor-analysis processes. The authors made explicit their hypotheses, described in detail their obtained results from testing the newly developed scale, and compared them with the “collegial nurse–physician relations” subscale from the nursing work index (NWI) and with the three subscales of the “attitudes toward healthcare teams scale” (ATHCTS). The authors reported on the obtained convergent, discriminant, and concurrent validity as well as on inter-item correlation (each of 15 hospitals) and inter-rater reliability (across hospitals) to test the stability of the data.

The IPC scale from Kenaszchuk et al. [7] was translated into German and Italian and further developed and adapted in collaboration with one of the original authors, who was the principal investigator of a research project in northern Italy [9]. The adapted German and Italian IPC scale has undergone validity and reliability testing with similar values to the original language validation (Cronbach’s alpha was 0.92 for the German and Italian versions) [10].

In line with Walters et al. [6], Peltonen et al. [8] suggested that item response theory should be applied as a new approach to test instrument properties and trait level measurement in individuals. Compared to analyses using classical test theory, item response theory offers several advantages, such as evaluating the contribution of each single item and people’s performance to the ratings. There are examples in the literature of the application of Rasch analysis (RA) to study the characteristics of instruments dedicated to the evaluation of interprofessional competence [11,12,13]. RA, named after the mathematician Georg Rasch, is an important exponent of the item response theory, offering an approach of mathematical modeling based upon a latent trait. RA transforms ordinal scales into interval measures [14,15]. Percentage units are converted into units that all have the same size resulting in a “logit,” short for “log odds unit”. They generally range from **−4** to **+4** and are centered at zero [16].

To the best of our knowledge, no study exists applying methods of item response theory regarding the IPC scale. Therefore, this study aimed to evaluate the validity and reliability of the German and Italian versions of the IPC scale from Kenaszchuk et al. [7], applying Rasch analysis as a method of item response theory to answer the following questions:How well do the single items of the IPC scale measure the perception of interprofessional collaboration? (Model fit/misfit)How is the person–item difficulty for the single items of the IPC scale distributed? (Person–item map or Wright map)Are there items particularly critical to measure the construct? (Item polarity)Does the single-item difficulty relate to the main characteristics of the sample (language version, gender, work experience, age-class, and workplace)? (Differential item analysis, hereafter DIF)Is the IPC scale able to distinguish between persons with lower and greater collaborative behavior? (Person separation index)

## 2. Materials and Methods

### 2.1. Study Design and Sample

Survey data from a northern Italian cross-sectional IPC study (2013–2016) were used. The IPC study gathered data using an online survey from seven different healthcare professions (nurses, physicians, dieticians, occupational therapists, physiotherapists, speech therapists, and psychologists) recruited from seven hospitals and 20 different community services of four health districts of one Health Trust in northern Italy, as published in previous articles [17].

From the original data set of 1532 nurses, 1182 nurses completed all items of the IPC scale (no missing values). We decided to apply Rasch analysis using the data of nurses as it is the professional group with the highest frequency of contact with physicians and with the strongest numerical presence in the IPC study. All other health professions were excluded from the analysis due to their smaller sample size and lower contact frequency.

### 2.2. Instrument and Data Collection

The multi-group measurement scale for IPC developed by Kenaszchuk et al. [7] was the result of a review of IPC measurement scales in which the authors did not find any scale with “multiple rater/target groups”. They therefore constructed and validated a new instrument composed of 13 statements rated by participants on a 4-point Likert scale (strongly disagree (1), disagree (2), agree (3), and strongly agree (4)), which were allocated to three key factors of IPC: that is, communication, accommodation, and isolation. The authors concluded from the results of their validation and reliability tests that the scale was suitable for use in assessing nurses’ collaboration with physicians.

For the Rasch analysis, data collected during the previous study using the IPC scale from nurses of both languages (Italian and German) were used. The reliability and validity values of the instrument in the two languages, Italian and German, were previously obtained and were consistent with the original English-language version [10]. Measurement invariance between the Italian and German versions was then assumed.

### 2.3. Data Analysis

The Rasch model [18] assumes that the probability of a given person–item interaction (in terms of rating high or low) is governed by the difficulty or simplicity of the item and the ability of the person: “A person having greater ability than another should have the greater probability of solving any item of the type in question and similarly, one item being more difficult than another one means that for any person the probability of solving the second item correctly is the greater one” [19]. Concurrently, the improbability of a person’s passing or failing a particular item is estimated item by item in terms of fit statistics. This is a comparison between what happened and what the model predicts should have happened based on the estimated measures.

**INFIT and OUTFIT** statistics are the most widely used diagnostic Rasch fit statistics. Comparison is made with an estimated value that is near to or far from the expected value. INFIT is more diagnostic when item measures are close to the person measures. OUTFIT is more diagnostic when item measures are far from the person measures.Observed average and outfit mean square values **(MNSQ)** were used to identify the compatibility of the data with the Rasch model.Uniform differential item functioning **(DIF)** was used to explore the stability of item difficulty to measure item invariance (item bias).The **Wright map or person–item map** [20,21] displays the results of the fits statistics and is delineated into two halves. The left side depicts the person measures and on the right side, the item measures, placed on the same ruler with indicated mean values and standard deviation for both.The reliability of the scale was evaluated by calculating **Cronbach’s alpha** coefficient and a **separation index for person and for items**. The reliability reporting how reproducible the person and item measure orders are (i.e., their locations on the continuum) is shown in Table 1, together with the reference values for the previous indices. Item separation index and item reliability are interpreted using the same criteria. According to Rasch guidelines, if the item reliability and separation are below the required values, a bigger sample is necessary; if the person reliability and separation are below the required values, the test requires more items [20,22].

All analyses were performed using the software Winsteps version 1.0.0 (Winsteps Rasch measurement computer program. Beaverton, OR, USA) [23]. For descriptive analysis, SPSS Statistics 18 (IBM Inc., Skokie, IL, USA) was used.

Item validity was measured according to point measure correlation (PTMea Corr.), INFIT and OUTFIT mean square (MNSQ), and standardized residual correlation.

## 3. Results

### 3.1. Sample Characteristics

Table 2 shows the distribution of the 1182 nurses by language version, gender, age, type of workplace, and work experience. A total of 1030 respondents were female (87.1%), and 53.2% of respondents were over 40 years old. Most of the participants worked in a hospital ward, and the mean numbers of years worked in healthcare and in the current ward/service were 18.0 and 10.9 years, respectively.

### 3.2. Reliability Measures and Separation Indexes

The results from the analysis of the IPC scale showed that the value of Cronbach’s alpha (α) is 0.92, which, according to George and Mallery [24], is rated as very good, indicating a high level of consistency. The separation person measure index is defined as N-groups with a significant difference in the value of the construct examined by the scale. A tool is considered more precise when it distinguishes between more groups of respondents. The minimum points of division are 2.0 [18]. For the adapted IPC scale, this index is 3.26, which is a good value according to Linacre [22]. The analysis identified four groups with significantly different ratings of the IPC, which demonstrates that the scale distinguishes among groups of persons with different collaborative attitudes. The item separation index is 22.23.

### 3.3. Item Polarity

Table 3 shows the values of the point measure correlation coefficient (PTMEA correlation) of the IPC scale obtained by Rasch analysis item by item. Examination of the item polarity tests the extent to which the developed construct achieves its goals and examines the relationship among the developed items of the respondents. According to Bond and Fox [18], to determine whether the item measured the construct, the value shown on the PTMea Corr. must be positive (+). According to the categorization of the correlation coefficient (absolute value of r), r ≤ 0.35 is generally considered a low correlation, between 0.36 and 0.67 is a moderate correlation, and between 0.68 and 1.0 is a high correlation.

As shown in Table 3, all values of PTMea Corr. are positive. This result indicates that all the items measure the construct, and no item needs to be excluded. Each item moves in one direction with other items that measure the construct. The correlation coefficients obtained for the IPC scale are strong (i.e., near/or > 0.7) for all items, except for items number 10 and 12 which have a moderate correlation.

### 3.4. Item Fit

Table 4 presents the INFIT and OUTFIT MNSQ values of item fit displayed from the lowest to the highest fit.

Statistical analysis was carried out to measure the suitability of items in order to identify items that should be greater than 0.5 and less than 1.5, according to Wright et al. [25]. If an item does not fulfill the requirements, then the results of the item (in our case, items 10 and 12) should be carefully evaluated.

Table 4 shows the values of INFIT MNSQ and OUTFIT MNSQ statistics of the IPC scale obtained for each item and the number of measured respondents (n = 1182). For this research, we used the total mean square INFIT and OUTFIT in the range proposed by Bond and Fox [18]. The item analysis showed that the mean square INFIT ranges from 0.68 to 1.83 and OUTFIT from 0.66 to 3.21. These results indicate that there are two items, number 10 and number 12, that exceeded the suggested limits.

### 3.5. Wright Map

Figure 1 shows the person–item map or Wright map that represents the results of the fits statistics.

The distribution of persons on the left side of the map shows a normal distribution. As can be seen in the person–item map (Figure 1), most of the items rank very closely to each other with no gap between them. Exceptions are items 10 and 12. Respondents perceived item 12 as much more difficult than the other items but considered item 10 much easier.

In this analysis, the means of items and of persons are not very close to each other. In addition, the simultaneous presence of multiple items on the same line indicates the possibility of redundancy, i.e., the inclusion in the scale of items that are very similar to each other. As Figure 1 demonstrates, this happens in three situations: items 3, 4, and 6; items 13, 8, and 9; and items 1 and 5 [26].

### 3.6. Item Difference (DIF)

Differential item functioning (DIF) was calculated for different groups of respondents according to gender (females and males), work experience (≤5 and >5 years), workplace (working on a ward or in a service), age (20–29, 30–39, 40–49, >50 years), and language (German and Italian). A noticeable DIF was defined according to two criteria: (1) DIF contrast >0.5 logits and (2) significance of the difference (t > 2.0) [22].

The results of the DIF analysis are shown in Table 5 and Figure 2. Regarding gender, a significant DIF was found for item 12, showing higher agreement among males. The same occurred for item 8 (considering age) for nurses 50 years old or older in comparison with nurses aged between 20 and 29 years. For the other variables, no significant DIF was found.

Additionally, other differences were observed, but they were not significant. Regarding items 8 and 10, answering seemed easier for nurses working in a service than for those working on a ward. For item 2, there was also a small difference in answers between German speakers and Italian speakers.

## 4. Discussion

The aim of this study was to explore the psychometric properties of the German and Italian versions of the adapted IPC scale, applying Rasch analysis as recommended by Walters [6] and Peltonen [8].

The results obtained in this study regarding reliability measures and separation indexes were good and confirmed the reliability of the scale in measuring interprofessional collaboration. This is in line with the reliability results in previous IPC studies [10,17] in which classical test theory was applied. The Rasch analysis revealed that four groups of respondents with different collaborative attitudes can be formed. This result underscores the reliability of the IPC scale because the more an instrument can distinguish between respondents, the more accurate it is [18].

The results obtained through item polarity analysis confirmed that all items of the scale measure the construct with high correlation values. For items 10 and 12, these values are slightly lower but remain positive. This indicates that no item is required to be deleted from the scale in regard to constructing validity.

The results concerning item fit expressed by total mean square INFIT and OUTFIT, shown in Table 4, do not reveal any critical issues regarding the individual items composing the scale, except for items 10 and 12. These exceed the suggested limits and should therefore be modified and improved to enhance the overall fit of the IPC scale.

The Wright map depicts the difficulty or ease the respondents had in agreeing with the items of the IPC scale. The distribution of respondents shown in Figure 1 on the left half of the map shows a normal distribution pattern. The right half of the map shows that item 12 (upper part), i.e., “the other profession thinks her/his work is more important than ours”, is found at the upper level of the map, far from the other items. This represents a greater difficulty and variability in answering that question for most respondents. By drawing an imaginary horizontal line at the level of this item, it is possible to estimate the number and number of people who responded to the item without any difficulty (adding the number of respondents in part above this line), which were 147 out of 1182 persons (12.4%), against 87.6% who found it difficult to answer. This item assesses an important issue within interprofessional collaboration. A prior study [17] discussed the idea that item 12 can provide insight into the connectedness (understood as reciprocal appreciation and recognition) between different professional groups. If connectedness is an expression of the readiness of the workforce for IPC, then it is important that this item remains on the scale. This item should not be modified as it indirectly allows us to draw conclusions on what the respondents think about the importance of their own profession. However, to better understand the complexity of this aspect, further research is necessary.

Item 10, that is, “important information is always passed on from us to the other profession”, is found in the opposite part of the Wright map. This item was the easiest item to agree with. The findings indicate that only a few people had difficulties in answering this item (15 out of 1182 persons, 1.3%). This is reflected by the very low variability in the answers to this item. The passing on of information concerning patients from one professional group to another seems to be an obvious and self-evident activity, but studies concerning patient safety reveal this as an aspect of weakness or risk, leading to adverse patient outcomes [27,28]. In measuring IPC between professional groups, we should assess information exchange behavior in professional groups other than the respondent’s own group. Item 10 only assesses the willingness or intrapersonal readiness (as described by Mallinsson et al. [29]) to communicate important information regarding patient care to another professional group and does not assess the characteristics of the Information Systems that can change in different workplaces. Therefore, asking respondents whether they pass the information on to another professional group may not be that informative in assessing the level of IPC. A suggestion could be modifying item 10 to differentiate between collaborative and uncollaborative health professionals in terms of information exchange. The item modification should consist of, at least, a simple reformulation of the item in a way that it assesses the perceived “incoming” information, meaning that the other professional group is passing important information about patient care to the respondent’s own professional group. Furthermore, the way in which item 10 is formulated can be easily influenced by social desirability and thus cause bias, in that instead of assessing how respondents pass on information at work, it instead may measure what they think is expected of them. A more profound change in this item would consist of a shift in its content: assessing “information sharing” would be more adequate for measuring interprofessional collaboration than assessing “passing information” (understood as a linear or unidirectional process), as also suggested by Anthoine et al. [30].

Comparing the responses according to the main characteristics of the respondents (item difference, DIF), some significant differences emerged (Table 5): in item 12, males expressed a higher agreement (*p* = 0.0015), while in item 8, the older nurses (>50) showed higher values (*p* = 0.0151) regarding “The other profession does usually ask for our opinions”. These results are in line with the findings of the previous analysis and confirm the stability of the instrument regarding different characteristics in the surveyed sample.

Peltonen et al. [8] criticized the myriad of different existing instruments with unclear rationales and often lacking in measurement properties. They recommend having at hand more generic instruments measuring IPC among different health professions and healthcare settings. The adapted IPC scale, also from the view of a non-classical test theory, fulfills this requirement with respect to nurses and physicians.

### Strengths and Limitations

The strength of this study is that it contributes to evaluating the measurement properties of the adapted IPC scale, utilizing an item response theory approach based on adequate sample size. Considering the characteristics of the sample and the instrument, the application of Rasch analysis seemed appropriate and effective for studying the psychometric properties of the IPC scale.

The IPC scale was created for the evaluation of one phenomenon among several professional groups. This study, as an initial attempt, applied Rasch analysis looking only into nurses’ perceptions of interprofessional collaboration in relation to physicians and not into others. It is, therefore, necessary to extend the analysis to other health professionals to validate the instrument as a multiple-rater scale.

## 5. Conclusions

The German and Italian versions of the IPC scale are useful for exploring interprofessional collaboration among nurses and physicians. However, they require further research and testing with other healthcare professions. By applying Rasch analysis, this study has shown that the IPC scale has good reliability and validity. The findings revealed that no item shows criticalities, except item 10, which requires modification to further improve the scale.

## Figures and Tables

**Figure 1 healthcare-10-02007-f001:**
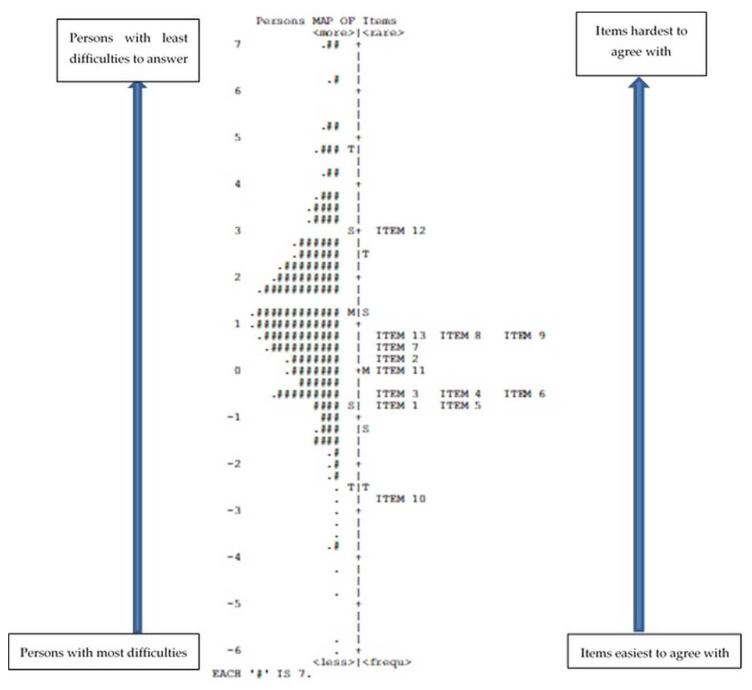
Person–item map of the adapted IPC scale. Notes: The line running vertically in the middle of the map divides the map into two halves. The left side shows the person measures, and the right side shows the item measures. The letters positioned on the ruler indicate the mean value (M), one standard deviation (S), and two standard deviations (T) for the persons and the 13 items of the IPC scale. The map allows us to evaluate how well the 13 items are distributed regarding the agreement level toward interprofessional collaboration. This evaluation is based on the distance between the mean item measure (“M” on the right side of the Wright map) and the mean person measure (“M” on the left side of the Wright map). The higher a person’s value is on the vertical line, the less difficulty they had agreeing with the item, while persons at the bottom had the greatest difficulty [21].

**Figure 2 healthcare-10-02007-f002:**
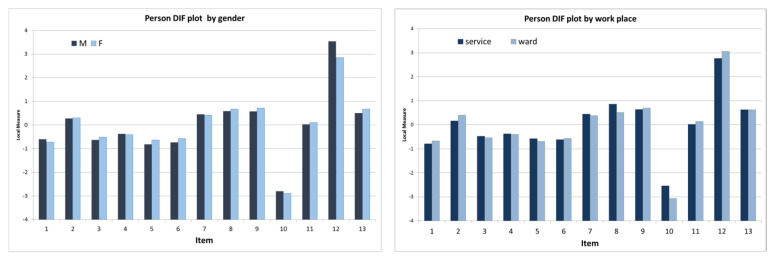
**Person DIF plots according to gender, work experience, workplace, age range, and language.** Notes: gender = F/M; age range = 20–29 years, 30–39 years, 40–49 years, 50+; workplace = working on a ward or in a service; work experience = ≤ 5 or >5 years; language = DE/IT.

**Table 1 healthcare-10-02007-t001:** Summary of analysis applied regarding item validity and reliability assessment.

Criteria	Statistical Index	Reference Values	
Person and item reliability	Person and item separation index (SE)	Separation index (SE) of ≥2.0 and reliability value ≥0.8 [18,22]	
Reliability of the scale	Cronbach’s alpha coefficient	Cronbach’s alpha	Internal consistency
		α ≥ 0.9	Excellent
		0.9 > α ≥ 0.8	Good
		0.8 > α ≥ 0.7	Acceptable
		0.7 > α ≥ 0.6	Questionable
		0.6 > α ≥ 0.5	Poor
		0.5 > α	Unacceptable
		[24]	
Item validity	Item polarity	Point measure correlation (PTMea Corr.) >0 [18]	
Item fit	Item dimensionality	Total mean square INFIT and OUTFIT within 0.5 to 1.5 [25]	
Item misfit	INFIT and OUTFIT mean square (MNSQ)	All items value ≥2.0 [18]	
Stability of item difficulty	Uniform differential item functioning (DIF)	(1) DIF contrast >0.5 logits, and (2) significance of the difference (t > 2.0) [22]	

**Table 2 healthcare-10-02007-t002:** Characteristics of the sample.

Characteristics	Number (*n* = 1182)	Percentage (%)
*Language*		
German	875	74.0
Italian	307	26.0
*Gender*		
Female	1030	87.1
Male	138	11.7
*Age (years)*		
20–29	173	14.6
30–39	373	31.6
40–49	435	36.8
50–59	188	15.9
>60	6	0.5
Not answered	7	0.6
*Workplace*		
Ward	599	50.7
Service	390	33.0
Not answered	193	16.3
*Work experience in the profession (years)*	Mean: 18.0	S.D. 9.8
*Work experience in the current ward/service (years)*	Mean: 10.9	S.D. 8.5

**Table 3 healthcare-10-02007-t003:** Item polarity.

Item Number	Item Description	PTMea Corr. Item
ITEM 10	Important information is always passed on from us to the other profession.	0.39
ITEM 12	The other profession thinks their work is more important than ours.	0.54
ITEM 1	We have a good understanding with the other profession about our respective responsibilities.	0.66
ITEM 4	The other profession and us share similar ideas about how to care for patients.	0.73
ITEM 13	The other profession is willing to discuss their new practices with us.	0.73
ITEM 9	The other profession is anticipating when we will need their help.	0.74
ITEM 3	I feel that patient care is adequately discussed between us and the other profession.	0.74
ITEM 11	Disagreement with the other profession is often resolved.	0.74
ITEM 2	The other profession is usually willing to take into account the convenience for us when planning their work.	0.75
ITEM 8	The other profession does usually ask for our opinions.	0.76
ITEM 5	The other profession is willing to discuss clinical issues with us.	0.76
ITEM 6	The other profession cooperates with the way we organize patient care.	0.76
ITEM 7	The other profession is willing to cooperate with us concerning new practices.	0.78

**Table 4 healthcare-10-02007-t004:** INFIT MNSQ and OUTFIT MNSQ statistics.

Entry Number	Item Logits Measures	Model S.E.M.	INFIT		OUTFIT		EXACT OBS%	MATCH EXP%
			*MNSQ*	*ZSTD*	*MNSQ*	*ZSTD*		
ITEM 10	−2.87	0.07	1.83	9.9	2.40	9.9	60.5	74.7
ITEM 12	2.93	0.05	1.80	9.9	3.21	9.9	52.4	61.4
ITEM 1	−0.70	0.06	0.93	−1.5	1.02	0.4	71.6	68.4
ITEM 4	−0.41	0.06	0.75	−6.3	0.76	−5.2	74.2	67.7
ITEM 13	0.64	0.05	0.94	−1.5	0.92	−1.7	70.0	64.7
ITEM 9	0.69	0.05	0.90	−2.3	0.91	−2.0	65.8	64.6
ITEM 3	−0.51	0.06	0.90	−2.4	0.88	−2.5	69.8	68.1
ITEM 11	0.09	0.05	0.88	−2.7	0.88	−2.6	68.9	66.6
ITEM 2	0.30	0.05	0.81	−4.6	0.81	−4.4	73.1	65.8
ITEM 8	0.67	0.05	0.91	−2.1	0.90	−2.2	64.5	64.7
ITEM 5	−0.66	0.06	0.95	−1.1	0.89	−2.3	68.6	68.3
ITEM 6	−0.59	0.06	0.71	−7.4	0.67	−7.4	76.8	68.1
ITEM 7	0.41	0.05	0.68	−8.4	0.66	−8.3	74.3	65.7
**MEAN**	**0.00**	**0.06**	**1.00**	**−1.6**	**1.15**	**−1.4**	**68.5**	**66.8**
S.D.	1.24	0.00	0.36	5.4	0.73	5.3	6.3	3.0

Note: INFIT and OUTFIT are reported as unstandardized MnSq or ZStandard (ZSTD) scores. INFIT and OUTFIT reported as a MnSq should have a value close to 1.0 with an acceptable range of 0.6–1.5 for the scales. The expected OUTFIT ZSTD value is 0, and values that exceed ±2 are interpreted as less than the expected fit to the model. Model underfit reduces the validity of the model and requires further investigation to determine the reason for the underfit. On the contrary, model overfit could result in a misinterpretation that the model worked better than expected, but it does not reduce the validity of the model. [18].

**Table 5 healthcare-10-02007-t005:** Differential item functioning results for the IPC scale.

Differential Item Functioning	IPC Scale
Gender	Item 12: higher agreement for males (0.69, *p* = 0.0015)
Language	No DIF
Work experience	No DIF
Workplace	No DIF
Age range	Item 8: higher agreement for nurses >50 vs. nurses 20–29 (0.56, *p* = 0.0151)

## Data Availability

Not applicable.

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
