# Peer review of "Evaluating Measurement Properties of the Adapted Interprofessional Collaboration Scale through Rasch Analysis"

_healthcare, 2022, doi:10.3390/healthcare10102007_

Round 1
Reviewer 1 Report
Given the push for more interprofessional collaboration in the healthcare arena, this paper covers a very important topic. An evidence-based measure for interprofessional collaboration is valuable for encouraging research into interprofessional collaboration. However, the paper needs a heavy edit and thorough copyedit. The English is awkward which makes it difficult to clearly assess what the authors are saying. The use of first person is something that I generally discourage in scientific papers. In addition, the reference style used varies. Sometimes only has last name of first author, other times has author name and date. One consistent reference style needs to be implemented throughout the paper. There are multiple places were the parenthesis after an author’s name are empty, publication dates need to be added.
Suggest adding the following reference, which describes utilization of Rasch analysis to evaluation the Interprofessional Education Collaborative Competence Self-Efficacy Tool: Axelsson M, Kottorp A, Carlson E, Gudmundsson P, Kumlien C, Jakobsson J. Translation and validation of the Swedish version of the IPECC-SET 9 item version. J Interprof Care. 2022 Feb 17:1-8. doi: 10.1080/13561820.2022.2034762. Epub ahead of print. PMID: 35175872.
Line 14: Not sure what you mean by “person-item measures”
Line 15: Define IPC in abstract.
Lines 17-18: Suggest changing “…Item polarity of all 13 items were positive indicating measuring well the construct. to ”…Item polarity of all 13 items was positive indicating good construct validity.” Is that what you mean to say?
Lines 26-27: IPC—the definition is collaboration of healthcare workers from two or more professions.
Lines 41, 46, 47: Kenaszchuk et al. references followed by empty parentheses. Did you intend to add publication date?
Line 46: Change underlined to emphasized.
Line 47: Change reporting to that reported.
Line 48: Delete “on item analysis for.”
Lines 66-67: Add reference for item response theory
Line 102: Empty parenthesis after authors names. Add publication dates.
Line 188: Empty parenthesis after authors names. Add publication dates.
Line 266: Empty parenthesis after authors names. Add publication dates.
Line 284: Wright Map is initial capital letters and italic; line 303 wright map is lower case letters. Make consistent.
Line 299: “In our opinion…” Instead of editorializing, explain the research findings.
Line 329: If the differences are significant, add p-value. If they are not statistically significant, use “considerable” in place of significant.
Line 339: Add the limitations of the Rasch analysis method. There are several articles published on the shortcomings of this methodology.
Line 350: The study evaluated collaboration between physicians and nurses. I don’t think you can claim you evaluated the tool with “different health professions.” Suggest changing to physicians and nurses and including the need for further research to test it with other health professions.
Author Response
Given the push for more interprofessional collaboration in the healthcare arena, this paper covers a very important topic. An evidence-based measure for interprofessional collaboration is valuable for encouraging research into interprofessional collaboration.
However, the paper needs a heavy edit and thorough copyedit.
Answer: Thank you for your recommendation. We have done this through a professional editing service.
The English is awkward which makes it difficult to clearly assess what the authors are saying.
Answer: Thank you for your comment. Here's a quick note on the process: We first made the content changes in the manuscript and then had it professionally proofread. We very much hope that the current version of the manuscript now reflects this and has also become more comprehensible.
The use of first person is something that I generally discourage in scientific papers.
Answer: We changed the whole text of the manuscript into a passive language style.
In addition, the reference style used varies. Sometimes only has last name of first author, other times has author name and date. One consistent reference style needs to be implemented throughout the paper. There are multiple places where the parenthesis after an author’s name are empty, publication dates need to be added.
Answer: Thank you for your attentive reading and feedback. We have now applied a consistent citation style as required.
Suggest adding the following reference, which describes utilization of Rasch analysis to evaluation the Interprofessional Education Collaborative Competence Self-Efficacy Tool: Axelsson M, Kottorp A, Carlson E, Gudmundsson P, Kumlien C, Jakobsson J. Translation and validation of the Swedish version of the IPECC-SET 9 item version. J Interprof Care. 2022 Feb 17:1-8. doi: 10.1080/13561820.2022.2034762. Epub ahead of print. PMID: 35175872. Answer: We added the suggested bibliography.
Line 14: Not sure what you mean by “person-item measures”.
Answer: Thank you for your comment. We have deleted this part because we have determined that it is not useful at this point and does not provide additional information.
Line 15: Define IPC in abstract.
Answer: We understood your comment to mean that we should write out the abbreviation so that it is clear what is meant.
Lines 17-18: Suggest changing “…Item polarity of all 13 items were positive indicating measuring well the construct. to ”…Item polarity of all 13 items was positive indicating good construct validity.” Is that what you mean to say?
Answer: Thank you for this suggestion. We applied it. It was very useful as it improves the clarity of the content.
Line 23 we added two more keywords as they help to specify better the content.
Lines 26-27: IPC—the definition is collaboration of healthcare workers from two or more professions.
Answer: Thank you for your comment. We added it as suggested.
Lines 41, 46, 47: Kenaszchuk et al. references followed by empty parentheses. Did you intend to add publication date?
Answer: Since we have changed the reference style, this problem no longer exists.
Line 46: Change underlined to emphasized.
Answer: we changed the term as suggested.
Line 47: Change reporting to that reported.
Answer: we changed the term as suggested.
Line 48: Delete “on item analysis for.”
Answer: We have deleted the sentence part as you suggested.
Lines 66-67: Add reference for item response theory.
Answer: Your suggestion helped us to reorganize and improve the whole paragraph (line 71-77). As you can see, we added the phrase regarding previous validity and reliability testing of the adapted IPC scale to an earlier paragraph to improve clarity and enhance the readability of the text. The sentence to which your comment was referred to was deleted as it no longer seemed necessary.
Line 102: Empty parenthesis after authors names. Add publication dates.
Answer: Since we have changed the reference style, this problem no longer exists.
Line 188: Empty parenthesis after authors names. Add publication dates.
Answer: Since we have changed the reference style, this problem no longer exists.
Line 266: Empty parenthesis after authors names. Add publication dates.
Answer: Since we have changed the reference style, this problem no longer exists.
Line 284: Wright Map is initial capital letters and italic; line 303 wright map is lower case letters. Make consistent.
Answer: Thank you for your attentive reading. We applied it now in a consistent manner.
Line 299: “In our opinion…” Instead of editorializing, explain the research findings.
Answer: We have omitted this part of the sentence because it has no added value in terms of content.
Line 329: If the differences are significant, add p-value. If they are not statistically significant, use “considerable” in place of significant.
Answer: We have provided the p-values in Table 6. Therefore, we have referred to them and added the values in the text.
Line 339: Add the limitations of the Rasch analysis method. There are several articles published on the shortcomings of this methodology.
Answer: We reorganized the limitation paragraph and made it more explicit. We also searched for studies applying Rasch analysis but found only reports on study limitations (sample size ...), but not on the methodology itself.
Line 350: The study evaluated collaboration between physicians and nurses. I don’t think you can claim you evaluated the tool with “different health professions.” Suggest changing to physicians and nurses and including the need for further research to test it with other health professions.
Answer: We fully agree with your comment and changed the text as suggested by you. Thank you very much.
Reviewer 2 Report
Revision of the article (healthcare-1880852):
Evaluating Measurement properties of the adapted Interprofessional Collaboration Scale applying a Rasch Analysis
This article aims to evaluate the validity and reliability of person-item measures of the adapted German and Italian version of the IPC scale applying a Rasch Analysis.
Recognizing the contribution of the study, however, there are important aspects to improve prior to assessing its possible publication:
· There should be no abbreviations in the abstract.
· Why did you decide to conduct the evaluation study of the IPC instrument only with nurses? Wouldn't it be more accurate to have included more categories of health professionals as you done in the mentioned previous study? It would be desirable if you explained this aspect in the article.
· If you make the following statement "we adapted this instrument in such a way that it could be applied to seven different professional groups", why was the study carried out only with nurses and not other types of health professionals? It would be important to include an explanation in this sense in the article.
· How can you reach the following conclusion if you have only done the analysis with a sample of nurses?: “The adapted IPC scale is useful to explore interprofessional collaboration among different health professions and allows assessing this phenomenon inside of health care organizations”.
· The bibliographic citations throughout the text do not follow the publication standards of the journal. The references section also does not follow the publication standards of the journal. Also, in line 41 you have also forgotten to include the bibliographic reference. In the reference of line 63 the same thing happens. This should have been checked before proceeding to submit the article to the journal. You can check the recommendations of the standards of the journal in the following link: https://www.mdpi.com/journal/healthcare/instructions
I hope these recommendations and suggestions help you to improve the article.
Kind regards
Author Response
There should be no abbreviations in the abstract.
Answer: We have adjusted the text so that no abbreviations are now included in the summary.
Why did you decide to conduct the evaluation study of the IPC instrument only with nurses? Wouldn't it be more accurate to have included more categories of health professionals as you done in the mentioned previous study? It would be desirable if you explained this aspect in the article.
Answer: Thank you for your question. We agree with you in principle, but since we wanted to check whether the Rasch analysis adds value (in terms of validity and reliability) to the IPC scale, we preferred to apply it to the largest group in the sample. We have specified and explained/described this in the text.
If you make the following statement "we adapted this instrument in such a way that it could be applied to seven different professional groups", why was the study carried out only with nurses and not other types of health professionals? It would be important to include an explanation in this sense in the article.
Answer: We have deleted this paragraph from section 2.2 because it was clear from your comment that it is not understandable that this part refers to the previous study. Also, we decided to delete it because it only creates confusion and does not add any value/content that is needed in this paper.
As you will note, we have added to 2.2 "and data collection" to the heading of this paragraph and inserted the last sentence of paragraph 2.1 into 2.2, since it refers to data collection. We have adjusted the wording of the sentence to improve clarity and understandability.
Your comment also helped us to hopefully make it much clearer in paragraph 2.1 why other professions were excluded.
How can you reach the following conclusion if you have only done the analysis with a sample of nurses?: “The adapted IPC scale is useful to explore interprofessional collaboration among different health professions and allows assessing this phenomenon inside of health care organizations”.
Answer: Thank you for your comment. We fully agree with you and have changed the text.
The bibliographic citations throughout the text do not follow the publication standards of the journal. The references section also does not follow the publication standards of the journal. Also, in line 41 you have also forgotten to include the bibliographic reference. In the reference of line 63 the same thing happens.
Answer: We checked the references and hopefully improved it throughout the whole paper. We also adapted the references to the citation style of the Journal.
This should have been checked before proceeding to submit the article to the journal.
Answer: Yes, you are right, but we had technical problems with incompatibility of the template and the version of our reference management program.
Round 2
Reviewer 1 Report
The manuscript is much improved. It now reads well. I found two small items:
Line 14: Add “of” after reliability.
Lines 131-136: Suggest moving this general information to the Introduction. Perhaps to line 75 after first sentence. And, move the second sentence on line 75 (starting with However) to next paragraph.
Author Response
Thank you for your comments. We changed the text as suggested.

Reviewer 2 Report
After reviewing the article by the authors, changes that improve it are appreciated. Likewise, the authors respond to most of the aspects of improvement identified by the reviewer previously. However, some minor changes still need to be applied in order to be accepted:
- Having included a new paragraph from line 75 to line 80, reference 14 is missing.
- In section “3.2. Reliability measures and Separation Indexes”, is passed from reference 24 to reference 27 (between lines 187 and 191). It is recommended to carry out a review of the entire citation of the bibliographic sources to guarantee that there is continuity in the numbering of the different bibliographic references used.
- Remember that in the abstract, the first time you write Interprofessional Collaboration Scale you must include (IPC) in parentheses if later in the abstract you want to use the abbreviation IPC.
- In the references section there are still important errors regarding the referencing of books and journal articles. Please see the Healthcare journal reference standards at the following link: https://www.mdpi.com/journal/healthcare/instructions
Author Response
After reviewing the article by the authors, changes that improve it are appreciated. Likewise, the authors respond to most of the aspects of improvement identified by the reviewer previously. However, some minor changes still need to be applied in order to be accepted:
- Having included a new paragraph from line 75 to line 80, reference 14 is missing.
Answer: we checked it and reordered the references in a continuous manner and marked in blue. Reference 14 has become now 17.
- In section “3.2. Reliability measures and Separation Indexes”, is passed from reference 24 to reference 27 (between lines 187 and 191). It is recommended to carry out a review of the entire citation of the bibliographic sources to guarantee that there is continuity in the numbering of the different bibliographic references used.
Answer : Thank you for your recommendation. We checked all citations and made sure that there is now continuity in numbering. Additionally we decided to remove reference 22: Souza MAP, Coster WJ, Mancini MC, Dutra F, Kramer J, Sampaio RF. Rasch analysis of the participation scale (P-scale): usefulness of the P-scale to a rehabilitation services network. BMC Public Health, 2017, 17,(1), pp. 934.
- Remember that in the abstract, the first time you write Interprofessional Collaboration Scale you must include (IPC) in parentheses if later in the abstract you want to use the abbreviation IPC.
Answer: we changed it accordingly.
- In the references section there are still important errors regarding the referencing of books and journal articles. Please see the Healthcare journal reference standards at the following link:
Answer: Thank you for the comment. We changed the style according to the journal reference standards.